# Pre-/-post-analyses of a feasibility study of a peer-based club intervention among people living with type 2 diabetes in Vietnam's rural communities

**Ngoc-Anh Thi Dang**[1]*, **Tuc Phong Vu**[1], **Tine M. Gammeltoft**[2], **Ib Christian Bygbjerg**[3], **Dan W. Meyrowitsch**[3], **Jens Søndergaard**[4]

1 Faculty of Public Health, Thai Binh University of Medicine and Pharmacy, Thai Binh, Vietnam, 2 Department of Anthropology, University of Copenhagen, Copenhagen K, Denmark, 3 Global Health Section, Department of Public Health, University of Copenhagen, Copenhagen K, Denmark, 4 The Research Unit for General Practice, Department of Public Health, University of Southern Denmark, Odense C, Denmark

* anhdtn@tbump.edu.vn

## Abstract

### Objectives

Insufficient self-management is a significant barrier for people with type 2 diabetes (T2D) to achieve glycemic control and consequently reduce the risk of acute and long-term diabetes complications, negatively affecting their quality of life and increasing their risk of diabetes-related death. This pre-post study aimed to evaluate whether a peer-based club intervention might reduce glycated hemoglobin from baseline to post-intervention and enhance self-management among people living with T2D in two rural communities in Vietnam.

### Methods

A pre-post study was implemented with 222 adults with T2D residing in two rural communities in Vietnam. We used a structured questionnaire, clinical examination, and glycated hemoglobin to evaluate the possible effects of a diabetes club intervention by comparing Glycated Hemoglobin (HbA1c), Body Mass Index (BMI), Blood Pressure (BP), and diabetes-related self-management behaviors at baseline and post-intervention. The data were analyzed using SPSS 20, applying two related sample tests (Wilcoxon and McNemar test) and a paired-sample t-test at a significance level of less than 0.05.

### Results

The findings indicated that after implementation of the intervention, there were no significant statistical differences when comparing pre-and post-intervention levels of the primary outcome HbA1c, but some components of diabetes self-management showed statistically significant improvement.

**Data Availability Statement:** All relevant data are within the paper and its Supporting information files.

**Funding:** Study was funded by the Ministry of Foreign Affairs of Denmark (DANIDA).

**Competing interests:** The authors have declared that no competing interests exist.

## Conclusions

After the peer support intervention in a Vietnamese rural community, there was no significant reduction in the primary outcome proportion of patients having an HbA1c less than 7%, but foot care knowledge and practice had improved.

## Trial registration

ClinicalTrials.gov NCT05602441.

## Introduction

Type 2 diabetes is recognized as a serious public health concern with a considerable impact on human life and health expenditures. A significant challenge for people living with type 2 diabetes (T2D) is inadequate self-management to maintain reasonable glycemic control and restrict cardiovascular risk factors such as hypertension and dyslipidemia, which increase the risk for diabetes complications, including diabetes foot complications, consequently accelerating morbidity, disability, and diabetes-related death risks [1, 2]. In spite of several significant diabetes medication innovations, many patients still struggle with diabetes care and control, which requires a multifaceted daily care continuum comprising a healthy diet, engagement in physical activities, blood glucose monitoring, foot care, and emotional coping [3, 4]. Moreover, individuals with T2D are currently at greater risk of being positive for COVID-19 [5], and inadequate blood glucose control has been proven to require more medical interventions, prolong recovery duration, and increase the risk of severe diseases after COVID-19 infection [6–8]. This may be attributed to high blood glucose, which facilitates the replication of the virus and weakens the immune system through lymphocytopenia. Poor blood glucose control also leads to vascular complications, harming organ function and worsening prognosis [9].

T2D has become a frequent non-communicable disease, which continues to increase in incidence and prevalence globally, particularly in low-income and middle-income countries [10, 11]. In 2021, an estimated 537 million adults, equivalent to 10.5% of the world's population aged 20–79, have diabetes, and 3 in 4 persons live in low- and middle-income countries (LMIC). In South-East Asia alone, it is predicted that the number of adults with diabetes will reach 113 million and 151 million by 2030 and 2045, respectively

Vietnam has been undergoing a rapid demographic transition with an aging population and a rising incidence of non-communicable diseases among the elderly, such as cardiovascular diseases, cancers, and diabetes, posting a burgeoning demand for the elderly's long-term healthcare. In line with other lower-middle-income developing countries globally, Vietnam's prevalence of diabetes has nearly doubled in just ten years [12]. Currently, Vietnam has up to 3.9 million people diagnosed with T2D, representing 6.1% of people aged 20–79 years, and the proportion of people with pre-or undiagnosed diabetes is assumed to be very high [13, 14]. It is estimated that the Vietnamese population's age-adjusted prevalence of diabetes (20–79 years) will increase to 7.1% by 2045 [14]. This situation poses challenges for Vietnam's long-term health services, where the healthcare workforce is currently understaffed, with 8.8 doctors per 10,000 inhabitants in 2019 [15] and having an imbalanced geographical distribution of the health workforce, especially highly specialized practitioner staff, between urban and rural areas [16]. Few community health centers at the primary care level provide diagnosis, treatment, or follow-up management of diabetes patients [17]. Instead, patients obtain medical care in high-

level hospitals, leading to overburdening of upper-level public hospitals. Moreover, overcrowding and excess workload in hospitals not only put pressure on the healthcare workforce [18] but also influence the quality of treatment and limit the time for a full consultation of daily self-management for patients [19]. A cross-sectional hospital-based study performed in Vietnam in 2015 showed that only 36% of patients with T2D met the treatment goal of glycated hemoglobin (HbA1c) < 7.0% (< 53 mmol/mol), as recommended by the American Diabetes Association Professional Practice Committee [20]. Furthermore, in the COVID-19 era, people living with T2D faced more self-management barriers and fear of being more vulnerable to COVID-19 due to underlying medical problems [21–23]. In the current context of Vietnam, informal support from non-professionals plays an essential role among people with T2D and would possibly assist with optimal diabetes self-management. This type of support aligns with the community-based primary healthcare trend to manage chronic disease globally, integrated into an empowerment framework [24–26]. Peer support models for diabetes management have long been referred to as the support between non-professional individuals who have common characteristics or concerns in diabetes. Such individuals can share and discuss experiential knowledge, identify, motivate, and facilitate good health behaviors, and alleviate pessimist thoughts and negative emotions, thus improving the capacity to overcome obstacles to self-care [27]. Adequate peer support has been recognized to improve health outcomes in many countries, especially in resource-constrained communities [25, 28–30]. An additional widely acknowledged type of supporter is the community health worker [24, 31, 32]. Community health workers, known as Village Health Workers (VHWs), are part of the grassroots public health care system in rural Vietnam, which belongs to a four-tiered health system structure, with widespread coverage from the central to the grassroots level, comprising central, provincial, district, and commune levels. VHWs serve as the frontline contact point for national healthcare services. They are engaged in multiple essential tasks in propagating and educating on health protection, environmental sanitation, food safety, primary healthcare measures, HIV/AIDS prevention and control, maternal and child health care, and family planning; first aid and primary medical care; detecting, monitoring, and reporting communicable and non-communicable diseases (NCDs) in the community; and other responsibilities [33]. Due to being closest to the residents and having a deep understanding of the local life, economic, and social situation, VHWs have demonstrated their ability to contribute significantly to NCD prevention programs [34–37]. Nevertheless, the NCD-related professional qualifications among most VHWs have not satisfied the population's needs, leading to lacking confidence in delivering preventive services to community members, especially those living with long-term chronic conditions [34–37]. Therefore, to boost the advantages and address the disadvantages of peer and VHW support for enhanced diabetes self-management capability in the rural communities, an intervention conducted in Vietnam's Thai Binh province as part of the larger interdisciplinary project, "Living Together with Chronic Disease: Informal Support for Diabetes Management in Vietnam" (VALID) (grant number 17-M09-KU), was developed by cooperation between the Thai Binh University of Medicine and Pharmacy (TBUMP), the University of Copenhagen, and University of Southern Denmark, funded by the Ministry of Foreign Affairs of Denmark (DANIDA) [38]. The project comprised an epidemiological component, an ethnographic component, and an intervention component. The intervention design was based on results from the epidemiological and the ethnographic research and further developed by people with T2D who participated in a participatory design workshop in 2019 [39].

The present pre-and post-study aimed to test the feasibility of a peer support intervention, emphasizing the possible effect of diabetes clubs to reduce HbA1C and improve T2D self-management among people living with T2D in two rural communities in Thai Binh Province, Vietnam.

## Materials and methods

### Study design

An intervention with a pre-post study was implemented in two rural communities in northern Vietnam. In January 2021, data on people living with T2D were collected prior to the implementation of the intervention. An intervention program, including "train-the-trainers" classes and diabetes clubs, took place monthly from March 2021 to January 2022. The post-intervention survey was carried out to collect data in January 2022.

### Setting

The study was conducted in Thai Binh Province, located 110 kilometers southeast of the capital of Hanoi. The province covers a natural area of 1542 km$^2$ and has a population of over 1.86 million people, with a predominantly agricultural economy of wet rice farming. The province is divided into one provincial city and seven districts, with a total of 241 communes, 10 sub-districts, and nine towns.

Vu Thu District is one of seven districts in Thai Binh Province, with 29 communes and one town. Two communes, namely Vu Hoi and Viet Thuan, with nine villages in each commune, were selected for the study.

In Vietnam, with health insurance coverage for about 90% of the population [40], people living with T2D enrolling in the health insurance scheme can access health care services at any district hospital or equivalent private hospital with a reduction of 80% - 100% depending on the subject, thereby avoiding out-of-pocket payments for health [41]. However, in 2021, Vietnam experienced several COVID-19 outbreaks nationwide, also affecting the study area in Thai Binh Province. Regardless of the effort of the health insurance scheme to cooperate with stakeholders to create the optimum conditions to ensure the benefits for health insurance participants, the epidemic still adversely influenced diabetes self-management. Due to fear of the epidemic or inability to arrange work caused by it, people with diabetes often omitted periodic routine check-ups, changed their medicine doses without consultation with a doctor, engaged in limited exercise, purchased out-of-pocket medication, and increased alcohol/beer use [42]. Moreover, due to two COVID-19 lockdowns, our intervention, including diabetes classes and clubs, had to be suspended in May and November 2021, respectively [43].

### The intervention overview

The research team organized a participatory design workshop in 2019. Participants were VHWs and people with T2D participating in the ethnographic study, along with their informal support persons. The aim was to develop a culturally appropriate intervention for diabetes self-management. In the qualitative study we identified obstacles, barriers and facilitations pertinent to having T2D and a strong desire for knowledge and peer support in self-management among people with T2D in rural communities [44, 45]. This knowledge fed into the process of designing the diabetes management intervention, "Living healthy and well with diabetes," including diabetes classes and diabetes clubs, to highlight two prominent aspects of diabetes self-management: education and peer support. Moreover, due to the limited availability of comprehensive and attractive health education communication materials on diabetes, we decided to develop a new set of diabetes educational materials, including 11 diabetes leaflets, to reach the elderly and those with limited reading skills easily. Subsequently, the leaflets were reviewed by experts in diverse fields, including internists, nutritionists, community health education and communications from TBUMP, and experts from the project advisory board. After pilot testing and modification based on the feedback from 63 people with T2D, all eleven

leaflets were written in clear and accessible language, with colorful illustrations, and printed in large font on glossy and thick A4-sized papers approved for printing by the Department of Culture and Information. These leaflets were delivered to all participants in the diabetes classes and clubs (S1 Table) [46].

During the participatory design workshop in 2019, the concept of club facilitators (CFs) was also coined, referring to those who lived with long-term T2D while being active in their communities and open about their disease. We requested VHWs from eighteen villages in two rural communities to identify their club's CFs to participate in the diabetes classes and assist VHWs in running the diabetes clubs monthly. Consequently, 30 VHWs and 32 CFs from eighteen villages in two rural communities participated in "train-the-trainer" classes held once a month, developed by the research team. The training topics focused on diabetes self-management principles and blood glucose fluctuations prevention and treatment on special occasions. In the classes, VHWs and CFs got opportunities to gain accurate diabetes knowledge, raise questions, be consulted, and discuss with the specialists from TBUMP to eliminate myths and attain good diabetes care behaviors, thereby building up their self-confidence when communicating and discussing with other members in the diabetes club.

Before and after each training of trainers session, VHWs and CFs organized and moderated interactive diabetes learning clubs with other patients in their villages who were willing to participate. The diabetes club meeting was held twice a month during a 9-month intervention, consisting of a focused delivery of the knowledge and discussion following the topic in the monthly class. Club meetings also included dedicated time for blood glucose testing and sharing experiences to deal with the difficulties of the periodic insurance examination process. Finally, the club meetings provided the participants with emotional support to deal with any stigma and sadness in their daily lives.

Core messages aimed to strengthen daily self-management and boost the spiritual bond between individuals with T2D toward a healthy, happy life with T2D. In addition, the intervention aimed to strengthen the capacities of grassroots healthcare workers—VHWs in diabetes consultations and communication in rural communities.

## Sampling

We estimated the sample size according to the formula:

$$n = \frac{\left[Z_{(1-\alpha/2)}\sqrt{2p\,(1-p)}+Z_{(1-\beta)}\sqrt{p_1(1-p_1)+p_2(1-p_2)}\right]^2}{(p_1-p_2)^2}$$

To compare the pre-post HbA1c proportion of statistical power of 80%, a two-sided level of significance of 5%, equal group sizes, the anticipated proportion in the baseline population of 65%, and to detect a difference in proportions of -0.15 between the two groups (test—reference group), at least 170 people should be included.

We recruited people diagnosed with T2D at age 40 years or above and residing in two selected communes diagnosed with or treated for T2D at district hospitals, private hospitals, and a provincial hospital. Based on the available list of people with T2D treated in the above hospitals, we employed a convenience sampling of all people with T2D who met the inclusion criteria and signed informed consent in two communes. Before starting the educational intervention program, a total of 311 participants participated in the pre-study, including a short clinical examination by internists and a face-to-face interview. After the intervention, the research team undertook follow-up assessments comprising quick clinical assessments and face-to-face re-interviews. Among 311 participants in the pre-study, seven (2.2%) had died,

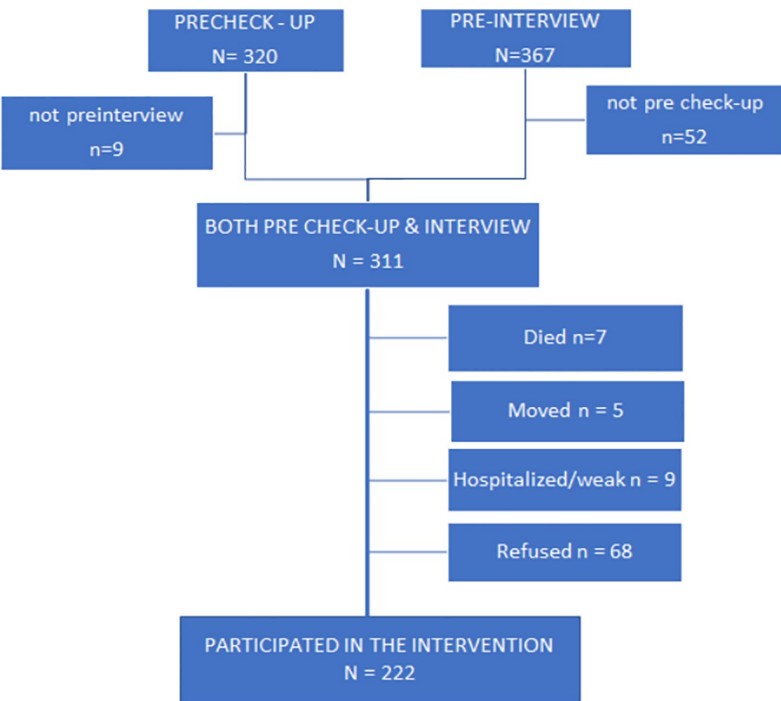

**Fig 1. Flow chart for recruitment and inclusion of study participants with T2D.**

nine (2.8%) were hospitalized or weak, 68 (21.4%) refused to participate (29 refused to check-up, and 39 refused to participate in the diabetes club intervention), and five (1.6%) had moved away by the period of data collection. Finally, 222 eligible individuals were recruited for the study, as shown in Fig 1.

## Inclusion criteria

Individuals diagnosed with T2D at age 40 years and above residing in two selected communes, without acute and/or severe illness, were able to answer the questionnaire completely, agreed to participate voluntarily, and participated in both pre–post-study and the intervention program.

## Exclusion criteria

Refusal to continue participating in the study, migration, or unforeseen reasons such as illness or hospitalization or not fulfilling other inclusion criteria.

## Ethics of research

The Medical Ethics Committee approved the ethical clearance of TBUMP, Vietnam (decision 11/2018, 23rd November 2018), registered at ClinicalTrials.gov (NCT05602441). Every participant was informed about the purpose of the study, that participation was voluntary, and signed the written consent before participating in the examination and interview. Participants could withdraw at any time from the study. The participants were checked-up at the commune healthcare centers and interviewed in their homes. The completed questionnaires were managed and stored securely at TBUMP.

## Data collection

**The interview part.** Data were collected using a paper-based structured and pilot-tested questionnaire. The questionnaire was in Vietnamese and pretested with people with T2D (N = 20) and validated to ensure the understandability and cultural relevance of the questions. The questionnaire comprised nine domains with 182 questions, including 1) the respondent's general information (7 questions); 2) the social network (8 questions); 3) the informal support (34 questions); 4) medication and use of health care services (21 questions); 5) mental health (38 questions, including SRQ-20 and DDS-17 scale); 6) quality of life (40 questions, including SF-36 scale and 1 question about happiness ladder); 7) self-management of diabetes and life-style regarding smoking, alcohol (12 questions); 8) being a member of diabetes club (22 questions), and 9) comments to the questionnaire. Only domains 1and 7 were used in the present study to describe the socio-demographics of respondents and compare diabetes self-management (S1 File).

Eight staff from TBUMP were trained to implement the interview at a 3-day workshop followed by a pilot with 20 people with T2D. Two interviewers conducted the pilot interview with the same 2–3 people with T2D separately. Subsequently, the results were compared. The conflicting reports were discussed and received feedback from the experts. In the field, the interviewers contacted VHWs to be directed to each patient's house in the village named on the list to conduct the interviews. Participants were informed about the project's purpose and asked to sign a consent form. Subsequently, the participants were interviewed face-to-face following the revised questionnaire.

**Glycated hemoglobin (HbA1c).** A laboratory technician took a blood sample for HbA1c testing using standard precautions from the participant's vein, then transported the blood samples in a cold container to the Provincial Center of Disease Control Laboratory for analysis on the same day. Even though the clinical goal for HbA1c level can differ depending on individual circumstances, a reasonable HbA1c goal for adults is 7% or less [1].

**Physical measurements.** To measure participants' height, they were asked to stand straight on a hard floor with their back and feet against the wall to which a height stature meter was attached, without footwear and headgear. Weight without footwear was recorded in kilograms. Body Mass Index (BMI) was calculated using the Asia-Pacific classification based on the formula of (weight (kg)/ [height $(m)^2$]).

After the participant sat relaxed with a comfortable back support chair for three to five minutes, the blood pressure (BP) and pulse readings were obtained with the left arm at the heart level using an automatic digital blood pressure monitor (Omron, Japan).

**Primary outcomes.** In our study, the primary outcome was the HbA1c index when comparing levels observed at pre- and post-intervention with an HbA1c treatment goal for adults equal to 7% or less.

**Secondary outcomes.** The secondary outcomes were changes in diabetes-related self-management practices, including awareness of hypo-/hyperglycemia, knowledge and practices of foot care, avoiding tobacco smoking, and low alcohol consumption. In addition, Physiologic and Anthropometric index changes, consisting of BMI, Systolic blood pressure (SBP), and Diastolic blood pressure (DBP), were measured.

## Analyses

The collected data were cleaned, including correcting spelling errors, handling missing data, and eliminating meaningless information before encryption. Data were entered into Epidata 3.1 and subsequently analyzed by Statistical Package for the Social Sciences (SPSS version 20.0) software. Descriptive statistics, including frequencies, percentages, means, and standard

deviations, were utilized to summarize the socio-demographic details of participants: age, gender, educational level, principal occupation, and the number of years of living with T2D. We determined the proportion of the respondents who met the recommended targets for HbA1c, BP, and BMI at pre-study and post-study and assessed significance levels by a McNemar test. We used the recommended HbA1c Hb1Ac <7%) [1] and BP targets (<140/90mmHg) [2] for adults with T2D proposed by American Diabetes Association, 2020. In this study, people with T2D were categorized into two glycemic control groups, consisting of good control (HbA1c <7%) and poor control (HbA1c ≥7%), and two blood pressure groups that did not reach the BP target (≥140 and/or ≥ 90mmHg) or reached BP target (<140/90mmHg). For BMI, we utilized the IDI & WPRO cut-off points for Asians: <23 kg/m$^2$ (classified as normal) and ≥23 kg/m$^2$ (classified as overweight/obesity) [47]. Paired-sample T-test was utilized to compare mean BMI and SBP at baseline versus after the education intervention. At the same time, we used Wilcoxon signed ranks 2-tailed test to find changes in median HbA1c and DBP, which follow a non-normal distribution.

The remaining variables, which comprised recognition of hypoglycemia or hyperglycemia signs and solutions, foot care practices, smoking, and alcohol drinking, were determined by the proportion of participants and comparing the difference at baseline and assessment by a McNemar test. When the patient had hypoglycemia, the correct responses consisted of going to the commune health centers, checking blood glucose, drinking fruit juice or milk, drinking sugar, eating candy, cakes, fruits, rice, instant noodle, and bread; while doing nothing, drinking pure water, or just resting without consuming carbohydrates was defined as an incorrect response. Regarding hyperglycemia, the patients' correct responses included going to the CHS, checking blood glucose, injecting insulin immediately, taking medicine, and drinking more pure water, while doing nothing was considered an incorrect response. Among smokers, we divided them into three groups, comprising light smokers (1–10 cigarettes per day), moderate smokers (11–20 cigarettes per day), and heavy smokers (more than 20 cigarettes per day) [48]. We used the standard drinks recommended from a systematic review and meta-analysis study conducted by Yuling Chen, which showed that ≤20 g of alcohol per day positively affected people with diabetes, equaling no more than two standard drinks per day [49]. Thus, there were two groups of alcohol consumption, including moderate alcohol consumption (<1–2 standard drinks per day) and over-daily fluid intake (≥3 standard drinks per day). For statistical significance, p-values of less than or equal to 5% were considered significant.

## Results

### Socio-demographic and health characteristics

The socio-demographic characteristics of the respondents are summarized in (Table 1). Out of 222 individuals living with T2D, 140 (63.1%) who had answered the questionnaire resided in the Vu Hoi commune, while 82 (36.9%) lived in the Viet Thuan commune. The proportion of female and male respondents was relatively equal, with 47.7% and 52.3%, respectively. The majority of respondents were in the age group 61–70 years (44.6%). People with T2D were predominantly farmers or unemployed/stayed at home (71.2%) and had educational levels from secondary school and below (78.4%). Approximately 40% of respondents had been diagnosed with T2D between one and five years, followed by those diagnosed with T2D over five to ten years (35.1%) and > ten years (23.4%).

### Glycated hemoglobin (HbA1c) changes

The change in the HbA1c indicator is indicated in Table 2. There was a significantly greater increase in median HbA1c after a 9-month intervention, 6.9% to 7.24%; difference = 0.34%

**Table 1. The socio-demographic characteristics of the participants of the VALID study 2019 (n = 222) (frequencies & percentages).**

| Demographic characteristics | Frequency (n) | Percentage (%) |
|---|---|---|
| **Commune** | | |
| Vu Hoi | 140 | 63.1 |
| Viet Thuan | 82 | 36.9 |
| **Gender** | | |
| Male | 116 | 52.3 |
| Female | 106 | 47.7 |
| **Age (years)** | | |
| **Mean ± SD** | | |
| 41–50 | 8 | 3.6 |
| 51–60 | 38 | 17.1 |
| 61–70 | 99 | 44.6 |
| 71–80 | 59 | 26.6 |
| >80 | 18 | 8.1 |
| **Educational level** | | |
| Primary school and below | 43 | 19.4 |
| Secondary school | 131 | 59.0 |
| High school | 38 | 17.1 |
| University/college | 10 | 4.5 |
| **Occupation** | | |
| Unemployed/ Stay at home | 66 | 29.8 |
| Farmer | 92 | 41.4 |
| Retired | 40 | 18.0 |
| Small trade/worker/government employee | 24 | 10.8 |
| **Marital status** | | |
| Married/living together | 171 | 77.0 |
| Single/Not living with spouse/ Divorced/Widowed | 51 | 23.0 |
| **Economic status** | | |
| Poor/ Near poor | 8 | 3.6 |
| Medium | 188 | 84.7 |
| Wealthy | 26 | 11.7 |
| **Years of living with diabetes** | | |
| >20 years | 6 | 2.7 |
| >10–20 years | 46 | 20.7 |
| >5–10 years | 78 | 35.1 |
| ≥1–5 years | 89 | 40.1 |
| < 1 year | 3 | 1.4 |
| **Total** | **222** | **100.0** |

(p < 0.001). However, there was no statistically significant difference between the proportion of participants with T2D reaching the treatment target for HbA1c when comparing pre-and post-intervention levels (p>0.05).

## Changing in self-management in the post-study

Table 3 indicates that the proportion of people with T2D who knew the signs of hypoglycemia after a 9-month intervention increased to 8.1%; the difference was insignificant. There was no

**Table 2. HbA1c in pre/post-study for people with T2D participating the intervention (n = 222).**

|  | Pre-study | | Post-study | | p-value |
|---|---|---|---|---|---|
|  | **n** | **%** | **n** | **%** |  |
| HbA1c[a]; % Median **(IQR)** | 6.9 (6.1–8.4) | | 7.24 (6.4–8.6) | | <0.001[b] |
| **Treatment target for HbA1c** | | | | | |
| Poor control (≥7%) | 108 | 48.6 | 120 | 54.1 | >0.05[c] |
| Good control (<7%) | 114 | 51.4 | 102 | 45.9 | |
| **Total** | **222** | **100** | **222** | **100** | |

[a]HbA1c: Glycated hemoglobin
[b]Wilcoxon Signed Rank test
IQR: interquartile range
[c]McNemar test

**Table 3. Knowing the signs and responses to hypo/hyperglycemia in pre/post-study for people with T2D participating in the intervention (n = 222).**

|  | Pre-study | | Post-study | | p-value[a] |
|---|---|---|---|---|---|
|  | **n** | **%** | **n** | **%** |  |
| **Hypoglycemia** | | | | | |
| *Knowing the signs* | | | | | |
| No | 105 | 47.3 | 87 | 39.2 | >0.05 |
| Yes | 117 | 52.7 | 135 | 60.8 | |
| *Frequency of hypoglycemia in the last month in perception* | | | | | |
| Total | 117 | 100 | 135 | 100 | |
| Never | 30 | 25.6 | 32 | 23.7 | >0.05 |
| 1–3 times | 61 | 52.1 | 84 | 62.2 | >0.05 |
| 4 times or more | 18 | 15.4 | 17 | 12.6 | >0.05 |
| Don't remember | 8 | 6.8 | 2 | 1.5 | >0.05 |
| *Response of the patients* | | | | | |
| Incorrect responses | 31 | 26.5 | 17 | 12.6 | >0.05 |
| Correct responses | 86 | 73.5 | 118 | 87.4 | |
| **Hyperglycemia** | | | | | |
| *Knowing the signs* | | | | | |
| No | 182 | 82.0 | 195 | 87.8 | >0.05 |
| Yes | 40 | 18.0 | 27 | 12.2 | |
| *Frequency of hyperglycemia in the last month in perception* | | | | | |
| Total | 40 | 100 | 27 | 100 | |
| Never | 6 | 15.0 | 11 | 40.7 | >0.05 |
| 1–3 times | 29 | 72.5 | 14 | 51.9 | >0.05 |
| 4 times or more | 5 | 12.5 | 2 | 7.4 | >0.05 |
| *Response of the patients* | | | | | |
| Incorrect responses | 27 | 67.5 | 7 | 25.9 | >0.05 |
| Correct responses | 13 | 32.5 | 20 | 74.1 | |

[a] McNemar test

significant difference in the proportion of correct hypoglycemia responses from 73.5% to 87.4% (p<0.05). Regarding hyperglycemia, there was no significant difference in the number of participants having correct hyperglycemia responses after the intervention compared to the pre-study, with 20 (74.1%) vs. 13 (32.5%).

There was a significant increase in knowledge and practice of foot care among 222 individuals with T2D., In the post-study, the proportion of patients knowing foot care methods exceeded the figure in the pre-study with 95.0% versus 81.5% (p<0.001). Not walking barefoot rose by 17.4% (p < 0.001); meanwhile, the McNemar test showed no significant difference in other foot care practices between the pre-study and the post-study (p>0.05). There was no significant difference in the ratio of current smokers among individuals living with T2D remained after the intervention (p<0.05). The number of smokers smoking over 20 cigarettes per day decreased from 7 (17.5%) to 3 (7.7%), without a significant difference compared to the pre-study. There was no significant difference in alcohol consumption with p > 0.05 (Table 4).

**Table 4. Self-management practices in pre/post-study for people with T2D participating the entervention (n = 222).**

| Self-management practices | Pre-study | | Post-study | | p-value[a] |
|---|---|---|---|---|---|
| | n | % | n | % | |
| **Know how to take care of foot** | | | | | |
| Total | 222 | 100 | 222 | 100 | |
| No | 41 | 18.5 | 11 | 5.0 | <0.001 |
| Yes | 181 | 81.5 | 211 | 95.0 | |
| **Foot care practices** | | | | | |
| Total | 181 | 100 | 211 | 100 | |
| Take care feet every day | 174 | 96.1 | 202 | 95.7 | >0.05 |
| Massages/ Acupuncture | 21 | 11.6 | 16 | 7.6 | >0.05 |
| Not walking barefoot | 111 | 61.3 | 166 | 78.7 | <0.001 |
| Soaking foot with herbs/salt | 56 | 30.9 | 55 | 26.1 | >0.05 |
| **Smoking** | | | | | |
| Total | 222 | 100 | 222 | 100 | |
| Non-smoker/Ex-smoker | 182 | 82.0 | 183 | 82.4 | >0.05 |
| Current smoker | 40 | 18.0 | 39 | 17.6 | |
| **Number of cigarettes per day** | | | | | |
| Total | 40 | 100 | 39 | 100 | |
| 1–10 cigarettes (light smokers) | 23 | 57.5 | 25 | 64.1 | >0.05 |
| 11–20 cigarettes (moderate smokers) | 10 | 25.0 | 11 | 28.2 | >0.05 |
| >20 cigarettes (high smokers) | 7 | 17.5 | 3 | 7.7 | >0.05 |
| **Drinking alcohol in the last month** | | | | | |
| Total | 222 | 100 | 222 | 100 | |
| Non-drinker | 155 | 69.8 | 153 | 68.9 | >0.05 |
| Current-drinker | 67 | 30.2 | 69 | 31.1 | |
| **Intake of a standard cup of alcohol per day among alcohol consumers** | | | | | |
| Total | 67 | 100 | 69 | 100 | |
| Moderate consumption (<1–2 cups) | 56 | 83.6 | 60 | 87.0 | >0.05 |
| Over daily fluid intake (≥3 cups) | 11 | 16.4 | 9 | 13.0 | |

[a] McNemar test

**Table 5. BMI and blood pressure means/medians in pre/post-study (n = 222).**

| Physiologic and anthropometric measures | Pre-study | Post-study | p-value |
|---|---|---|---|
| BMI*; kg/m$^2$ (**Mean ± SD**) | 22.7 ± 3.1 | 23.0 ± 3.1 | <0.05 (0.019) |
| SBP*; mmHg (**Mean ± SD**) | 142.5 ± 21.0 | 142.7 ± 21.2 | >0.05 (0.869) |
| DBP **; mmHg **Median (IQR)** | 82 (75.0–90.0) | 80 (73.0–89.0) | <0.001 |

*Paired t-test

** Wilcoxon Signed Rank test

SD: Std Deviation; IQR: interquartile range

**Table 6. The proportion of patients reaching the treatment target for BMI and blood pressure in pre/post-study (n = 222).**

| Physiologic and anthropometric measures | Pre-study | | Post-study | | p-value[1] |
|---|---|---|---|---|---|
| | n | % | n | % | |
| **BMI** | | | | | |
| Normal (<23kg/m$^2$) | 119 | 53.6 | 107 | 48.2 | >0.05 |
| Overweight/Obesity (≥23 kg/m$^2$) | 103 | 46.4 | 115 | 51.8 | |
| **SBP** | | | | | |
| <140mmHg | 98 | 44.1 | 96 | 43.0 | >0.05 |
| ≥140mmHg | 124 | 55.9 | 126 | 56.8 | |
| **DBP** | | | | | |
| <90mmHg | 144 | 64.9 | 168 | 75.7 | <0.05 (0.004) |
| ≥90mmHg | 78 | 35.1 | 54 | 24.3 | |
| **Blood pressure groups** | | | | | |
| Did not reach BP target (≥140 and/or ≥ 90mmHg) | 139 | 62.6 | 129 | 58.1 | >0.05 |
| Reach BP target (<140/90mmHg) | 83 | 37.4 | 93 | 41.9 | |
| **Total** | **222** | **100** | **222** | **100** | |

[1] McNemar test

## Physiologic and anthropometric changes

The changes in physiologic and anthropometric indicators are indicated in Table 5. There was a significantly greater increase in mean BMI after a 9-month intervention, 22.7 ± 3.1 kg/m$^2$ to 23.0 ± 3.1 kg/m$^2$; difference = 0.3 (p = 0.019). DBP was illustrated as a significantly greater reduction in the median with a difference equal to -2.0mmHg (p < 0.001). There was no significant change in mean systolic blood pressure.

Table 6 illustrates that there was no statistically significant difference between the proportion of patients reaching the treatment target for physiologic and anthropometric indicators, except for DBP indicators, with the rate of those reaching the treating target (DBP < <90mmHg) increased from 64.9% in the pre-study to 75.7% in the post-study (p<0.05).

## Discussion

This pre-/-post-study aimed to assess the feasibility of a peer-based club intervention in influencing the HbA1c index and self-management among people living with T2D in rural communities. Albeit there were no effects on the primary outcome, the study still shows promising results that should be addressed in a randomized controlled study. A total of 39 persons refused to participate in the intervention. We have not systematically recorded the reasons for

this nonparticipation, but believe that there is a pletphora of reasons including logistics, reluctance to partipate in new types of interventions and the COVID-19 crisis probably also contributed our intervention participants' dropout. Future studies should aaddress how to reach out to these nonparticipants in order to improve participation.

## Glycated hemoglobin (HbA1C) changes

The average HbA1C levels demonstrated an unexpected statistically significant though minimal and probably not clinically relevant increase from 6.9 to 7.24% from pre-intervention to post-intervention. Our result contrasts with some previous studies in Cameroon [50], Uganda [25], and Guatemala [24], where peer support significantly improves HbA1c in people with T2D. One explanation for the slight increase in HbA1c was that our intervention was conducted during the delta variant outbreak of COVID-19 in Vietnam [43], not sparing Thai Binh Province. Albeit it is unknown how many of the participants were infected by COVID-19 but nationwide social distancing during COVID-19 (45–50) posed challenges like a more sedentary lifestyle and consequently inadequate diabetes self-management among people with T2D [21]. Individuals with T2D also feared being more susceptible to COVID-19 infection due to comorbidity [21–23], leading to the delay or avoidance of regular check-ups at healthcare facilities [51–55]. Further, we do not know the changes in HbA1C among patients who did not participate in our study. Another explanation for the small increase in HbA1c is that T2D is a progressive condition [56]. In our study, the elderly predominated with 79.3% of respondents aged 60 and above; thus, although the normal patterns of development of T2D can be slowed down based on good self-management, they may still worsen with time. In addition, we might hypothesize that having less fear of diabetes complications may contribute to the small increase HbA1c index due to a slightly less adherence to recommendations. However, our data does not support this assumption and future analyses should look into this area. We also found similarities in the previous studies, which exhibited a non-significant difference in HbA1c level [29–31] after the intervention. Moreover, the number of studies with similar findings was limited, which may be caused by the fact that positive results tend to be published, while no results or negative tend to be unpublished [57].

## Self-management behaviors

In the context of a healthcare workforce deficiency and overcrowding in Vietnam's hospital system, our intervention indicated notable contributions of the peer-based club model to the change in diabetes self-management behaviors among individuals with T2D in rural communities. Despite hypoglycemia being a dangerous acute complication of diabetes, leading to unfavorable health effects and even mortality [58], people living with T2D often pay little attention to this complication [59–61]. After the intervention, although without statistical significance, the proportion of participants who recognized the signs of hypoglycemia and had correct responses to hypoglycemia increased compared to the pre-study, to 8.1% and 13.9%, respectively, which could suggest an effect of the intervention in raising hypoglycemia awareness to reduce and prevent adverse health effects among people living with T2D. Unfortunately the improvement in self-management behaviors was not reflected in improved HbA1c.

In addition, over time, having diabetes, especially poorly controlled diabetes, has led to a higher risk of developing foot complications due to nerve damage and poor circulation. Foot problems are one of the most common diabetes-related chronic complications, which may become severe and lead to amputations and even mortality if treated improperly [62]. Therefore, individuals with T2D need to be provided adequate foot care knowledge to enhance awareness and attain good behaviors, preventing or delaying these unexpected diabetes-related

foot problems. Using appropriate footwear is one of the strategies to prevent diabetic foot complications [62]. In contrast, several previous studies detected that walking barefoot is common behavior in developing countries and a noticeable risk factor leading to the high prevalence of diabetes foot problems [63–65]. Our study showed an improvement in knowledge and reduced the proportion of walking barefoot behavior among people with T2D having peer support, with a significant increase of 17.4% of respondents not walking barefoot (p < 0.001). Our findings were similar to the results of previous studies, which pointed out that self-care activities, including foot care and diet adherence, improved among people with T2D who engaged in peer support [28, 66, 67]. The effectiveness of community-based peer support intervention was also demonstrated in Assah et al.'s study in Cameroon, with a significant contribution to self-care behaviors improvement, comprising general diet, exercise, blood glucose test, and foot score [50]. The considerable improvement in self-care behavior—not walking barefoot, suggests an effect of awareness reinforcement among people with diabetes via teaching classes and delivering and sharing knowledge and experiences at peer-based diabetes clubs. Despite the remaining self-management behaviors, including hyperglycemia control, smoking, and alcohol consumption, did not indicate a statistically significant difference, the improvement rate was remarkable despite COVID-19's negative effect. Therefore, this intervention model requires more elaborate studies in different contexts and settings and long-term follow-up after implementation since T2D is a condition that is likely to have to live lifelong and affects the patient, the family, and the community.

### Physiologic and anthropometric changes

Another unexpected result was a significant increase in mean BMI among people with T2D after a 9-month intervention, from $22.7 \pm 3.1$ kg/m$^2$ to $23.0 \pm 3.1$ kg/m$^2$; difference = 0.3 kg/m$^2$ (p = 0.019). Although we conducted pilot testing and modified the educational materials, the strict application of the information provided in the educational materials in diabetes patients' daily lives seems quite hard, depending on each individual's diabetes and life status, especially during the COVID-19 lockdown. Several studies on lifestyle and dietary habits change showed a remarkable increase in consumption of instant meals and junk foods while dropping fruits and vegetables [68] or an increase in sedentary behaviors [69, 70] during the COVID-19 era. These findings also were observed for gaining weight and poor BMI control.

Although the mean DBP in the pre-study dropped from 82 to 80 mmHg post-study (p < 0.001), the change was modest and not clinically relevant. Among the participants, 35.1% had a DBP exceeding 90 mmHg in the pre-study compared to around 24% in the post-study. The rate of people with T2D having blood pressure reaching the treatment target (<140/90mmHg) increased from 37.4% to 41.9%. Improvement of DBP was reported in the study of Baumann et al. in 2014 among Ugandan adults with diabetes with a clinically reduce of 9.12mmHg of DBP compared to pre-intervention [25]. Another survey by F. K. Assah et al. found a similar drop of 6 mmHg of DBP [50].

### Strengths and limitations

To our knowledge, our study is the first to apply a peer-based club intervention model for people with T2D in rural Vietnam. Therefore, the strength of this study is that it provides a new picture of the impacts and the feasibility of a peer-based club intervention model to improve self-management among people living with T2D in rural communities in Vietnam. Our findings have opened up potential opportunities for more elaborate studies in different contexts and settings and long-term follow-up to fully evaluate the positive effects of this model on diabetes self-management in other rural areas in Vietnam.

There are some limitations to this study. Firstly, the sample size for the study was smaller than needed according to our power calculation, due to the dropout of the participants in the COVID-19 period, which may affect the statistically different comparison of some variables. Secondly, the validity of the outcome measures. For example, we chose the target treatment of HbA1c and BMI for Asians to be under 7% and $<23 \text{kg/m}^2$, respectively, which may differ from western studies, limiting the generalization of our results to other research. Another limitation was that the way clubs and classes were undertaken might not be optimal for achieving the goals of the significant intervention change in the HbA1c index; rather, it was more aligned with changing self-management behaviors among people with T2D. Lastly, this study was not a randomized controlled trial; thus, we cannot rule out that some bias may have influenced our results. In contrast, it has created favorable opportunities for further studies based on our findings.

## Conclusion

This study indicated that the intervention had no clinically significant beneficial effects on the primary outcome of HbA1c. However, our findings suggested that peer support in the diabetes club intervention enhanced aspects of diabetes self-management in rural communities in Vietnam. Despite these relatively modest effects that probably were heavily influenced by the COVID-19 restrictions with the influence of lifestyle and challenges in conducting the intervention as planned, we suggest on the basis of our findings that the concept of diabetes clubs should be elaborated and a randomized controlled trial should be conducted to determine effects on primary and second outcomes. Our study has shown that our intervention would be feasible and have positive effects even in very hard times.

## Supporting information

**S1 Table. Eleven leaflets used as educational materials in diabetes classes and clubs.**
(DOCX)

**S1 Checklist. STROBE statement—Checklist of items that should be included in reports of observational studies.**
(DOCX)

**S1 File. The questionnaire used in the paper.**
(SAV)

**S2 File. Data of the study.**
(DOCX)

**S3 File.**
(SAV)

## Acknowledgments

We are grateful to the teachers from TBUMP, Assoc. Prof. Ngô Thị Nhu, Assoc. Prof. Ninh Thị Nhung, Ph.D. Vũ Thanh Bình, MD. Lê Minh Hiếu, MPH. Nguyễn Thị Ái, MPH. Vũ Thị Kim Dung, and MPH. Đặng Thị Thu Ngà participated in the teaching and observation tasks in the diabetes classes and clubs during the intervention. Furthermore, we would like to express our gratitude to Dr. Phạm Tất Thắng and Dr. Trần Xuân Hi, leaders of Vu Hoi and Viet Thuan commune healthcare center, the commune health workers, and 30 village health workers from the two communes in Vu Thu district, Thai Binh province for supporting the

intervention. Finally, we thank those living with T2D from Vu Hoi and Viet Thuan communes who participated in the pre-/-post-study.

## Author Contributions

**Conceptualization:** Ngoc-Anh Thi Dang, Tine M. Gammeltoft, Ib Christian Bygbjerg, Jens Søndergaard.

**Data curation:** Ngoc-Anh Thi Dang, Tuc Phong Vu, Tine M. Gammeltoft, Ib Christian Bygbjerg, Jens Søndergaard.

**Formal analysis:** Ngoc-Anh Thi Dang.

**Methodology:** Ngoc-Anh Thi Dang, Dan W. Meyrowitsch.

**Project administration:** Tine M. Gammeltoft.

**Supervision:** Ib Christian Bygbjerg, Dan W. Meyrowitsch, Jens Søndergaard.

**Writing – original draft:** Ngoc-Anh Thi Dang.

**Writing – review & editing:** Ngoc-Anh Thi Dang, Tuc Phong Vu, Tine M. Gammeltoft, Ib Christian Bygbjerg, Dan W. Meyrowitsch, Jens Søndergaard.

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
