## [Decision Letter · Decision Letter 0]

19 Jun 2023

PONE-D-23-09542Pre-/-post-analyses of a  feasibility study of a peer-based club intervention among people living with type 2 diabetes in Vietnam’s rural communitiesPLOS ONE

Dear Dr. Dang,

Thank you for submitting your manuscript to PLOS ONE. After careful consideration, we feel that it has merit but does not fully meet PLOS ONE’s publication criteria as it currently stands. Therefore, we invite you to submit a revised version of the manuscript that addresses the points raised during the review process.

We look forward to receiving your revised manuscript.

Kind regards,

Tahir Turk, PhD

Academic Editor

PLOS ONE

Journal Requirements:

“The present study is part of the interdisciplinary research project, Living Together with Chronic Disease: Informal Support for Diabetes Management in Vietnam (VALID) (17-M09-KU), funded by the Ministry of Foreign Affairs of Denmark (DANIDA).”

“Study was funded by the Ministry of Foreign Affairs of Denmark (DANIDA)”

Additional Editor Comments (if provided):

Reviewer 1. Accepted with no revisions.

Reviewer 2 Comments: This is an interesting study. I have some comments on some aspects of the manuscript.

In the introduction, there is no information on the global prevalence of diabetes. It is useful to give the global picture before the burden of the disease in Vietnam

On page 6 under the Intervention overview, authors mentioned a previous qualitative study on line 157 but information has not been provided on how this is linked with the current study. It will be useful to provide this information

Materials and Methods

Page. 10. Under sampling, 39 people in the pre-study refused to participate in the diabetes club intervention but mention has been made of this as a possible influencing factor of the outcome/results. Indicate the reasons why they refused to participate in the intervention. That may help the explanation.

Results

Combine Table 4 and 5 into one table as these are the self-management practices

Combine table 6 and 7 into one table

Discussion

All the sub-sections under the discussion have been linked with the COVID-19 pandemic but there is not report on whether any of the participants suffered the COVID-19 It will be appropriate to mention this in order to put the discussion in the appropriate context as mentioned under the settings on page 7 line 145 - 154

For the discussion of the HbA1c changes, COVID-19 is used as a plausible reason for no significant difference but this did not seem to affect self-care practices including medication adherence. Self-care practices are known to have effect on the HbA1c levels. it will be useful to find explanations for the improvement in self-care practices but didn't reflect in the HbA1c

On page 22 line 432 - 434, a possible reason for increase in HbA1c is indicated as less fear of diabetes complication. Support this statement with evidence

Reviewers' comments:

Reviewer's Responses to Questions

**Comments to the Author**

1. Is the manuscript technically sound, and do the data support the conclusions?

Reviewer #1: Yes

Reviewer #2: Yes

2. Has the statistical analysis been performed appropriately and rigorously? 

Reviewer #1: Yes

Reviewer #2: Yes

3. Have the authors made all data underlying the findings in their manuscript fully available?

Reviewer #1: Yes

Reviewer #2: No

4. Is the manuscript presented in an intelligible fashion and written in standard English?

Reviewer #1: Yes

Reviewer #2: Yes

5. Review Comments to the Author

Reviewer #1: The topic is interesting and relevant to the present day as there has been a rapid increase in diabetes-related disease burden in most Asian countries and there is a dire need for approaches to treatment and care in diabetes management. Peer support group, use of positive deviant people within the community, use of lady health workers within the effected community intervention strategy has been employed previously and has shown good result for NCD and other lifestyle behaviours change. This is more useful particularly in resource-constrained rural communities where informal health support through community members or peer-based club intervention is a possible option for improving the quality of life for people living with diabetes. I enjoyed reading the paper which was well written with proper English and scientifically well laid out. Moreover, the results were negative in terms of primary objectives but the authors have published their results giving valid explanation.

Introduction: good & comprehensive and recent literature incorporated. Was not sure if the introduction can have sub headings, as the author has 2-3 subheadings within the intro section.

Methods: very well presented, however if the author can specify the name of the organization within the paper about the cutoffs for HbA1c goal for adults is 7 % or less, or for blood pressure so the reader can know how the author has decided on these cutoffs instead of looking at the references. She has done for other variables like BMI.

Fig 1. The flow chart is blurred, and needs to be improved.

Results: The authors table titles needs to be refined for example Table 1 title is Table 1. The respondent’s socio-demographic (n = 222). For example, A more appropriate title could be:

Table 1. The socio-demographic characteristics of the participants of the VALID study 2019 (n 222) (Frequencies & percentages). I suggest the author reviews all table title and write it more scientifically. Also some tables may need additional footnotes instead of only statistical test name.

Discussion: Does the Plos 1journal allow subsection in the discussion section? as currently the author has subsections with the discussion part. Otherwise well composed.

References:

suggest to go through these recent articles on similar topics:

1. Meyrowitsch DW, Nielsen J, Bygbjerg IC, Søndergaard J, Thi DK, Huyen DBT, Gammeltoft T, Duc TN

Unmet needs for informal care among people with type 2 diabetes in rural communities in Vietnam.

Public health in practice (Oxford, England). 2023.

2. Li S, Yin Z, Lesser J, Li C, Choi BY, Parra-Medina D, Flores B, Dennis B, Wang J

Community Health Worker-Led mHealth-Enabled Diabetes Self-management Education and Support Intervention in Rural Latino Adults: Single-Arm Feasibility Trial.

JMIR diabetes. 2022.

3. Development and evaluation of self-care intervention to improve self-care practices among people living with type 2 diabetes mellitus: a mixed-methods study protocol.

BMJ open. 2021.

4. Shiyanbola OO, Maurer M, Mott M, Schwerer L, Sarkarati N, Sharp LK, Ward E

A feasibility pilot trial of a peer-support educational behavioral intervention to improve diabetes medication adherence in African Americans.

Pilot and feasibility studies. 2022

Reviewer #2: This is an interesting study. I have some comments on some aspects of the manuscript.

In the introduction, there is no information on the global prevalence of diabetes. It is useful to give the global picture before the burden of the disease in Vietnam

On page 6 under the Intervention overview, authors mentioned a previous qualitative study on line 157 but information has not been provided on how this is linked with the current study. It will be useful to provide this information

Materials and Methods

Page. 10. Under sampling, 39 people in the pre-study refused to participate in the diabetes club intervention but mention has been made of this as a possible influencing factor of the outcome/results. Indicate the reasons why they refused to participate in the intervention. That may help the explanation.

Results

Combine Table 4 and 5 into one table as these are the self-management practices

Combine table 6 and 7 into one table

Discussion

All the sub-sections under the discussion have been linked with the COVID-19 pandemic but there is not report on whether any of the participants suffered the COVID-19 It will be appropriate to mentbionthis in order to put the discussion in the appropriate context as mentioned under the settings on page 7 line 145 - 154

For the discussion of the HbA1c changes, COVID-19 is used as a plausible reason for no significant difference but this did not seem to affect self-care practices including medication adherence. Self-care practices are known to have effect on the HbA1c levels. it will be useful to find explanations for the improvement in self-care practices but didn't reflect in the HbA1c

On page 22 line 432 - 434, a possible reason for increase in HbA1c is indicated as less fear of diabetes complication. Support this statement with evidence

6. PLOS authors have the option to publish the peer review history of their article (what does this mean?). If published, this will include your full peer review and any attached files.

Reviewer #1: No

Reviewer #2: No

---

## [Author Response · Author response to Decision Letter 0]

2 Aug 2023

Dr. Tahir Turk

Academic Editor

PLOS ONE

Dear Editor, dr. Tahir Turk

We would like to thank you and your reviewers for your thorough review of our manuscript “Pre-/-post-analyses of a feasibility study of a peer-based club intervention among people living with type 2 diabetes in Vietnam’s rural communities” (Manuscript ID PONE- D-23-09542) and for giving us the opportunity to revise it. We believe that this process has significantly improved our manuscript. A point-by-point response can be found below. Changes in the manuscript have been highlighted in italics.

All authors have approved the contents of this manuscript as well as the revisions that we have made.

Thank you for your continued consideration and the opportunity to publish in Plos One.

Sincerely, 

Ngoc-Anh Thi Dang, MPH

Department of Public Health

Thai Binh University of Medicine and Pharmacy 

Thai Binh, Vietnam 

 

Journal Requirments

Please remove any funding-related text from the manuscript and let us know how you would like to update your Funding Statement. 

We have deleted this part from the acknowledgment section and added a section to the manuscript:

“Funding

The study is funded by the Ministry of Foreign Affairs in Denmark (DANIDA) Grant no 17-M09-KU“

Reviewer #1: 

Accepted with no revisions.

Thank you

1. Introduction: good & comprehensive and recent literature incorporated. Was not sure if the introduction can have sub headings, as the author has 2-3 subheadings within the intro section.

Thank you. We have deleted the subheadings in the introduction section. Please see in the manuscript. 

2. Methods: very well presented, however if the author can specify the name of the organization within the paper about the cutoffs for HbA1c goal for adults is 7 % or less, or for blood pressure so the reader can know how the author has decided on these cutoffs instead of looking at the references. She has done for other variables like BMI.

We have added to the manuscript:

“We used the recommended HbA1c Hb1Ac <7%) (1) and BP targets (<140/90mmHg) (2) for adults with T2D proposed by American Diabetes Association, 2020”

3. Fig 1. The flow chart is blurred, and needs to be improved.

Thank you. We have updated a new flow chart 

4. Results: The authors table titles needs to be refined 

Thank you. We have changed the titles of the tables

Reviewer #2:

1. In the introduction, there is no information on the global prevalence of diabetes. It is useful to give the global picture before the burden of the disease in Vietnam.

Thank you. We have added in line 67 -72:

“T2D has become a frequent non-communicable disease, which continues to increase in incidence and prevalence globally, particularly in low-income and middle-income countries (3, 4). In 2021, an estimated 537 million adults, equivalent to 10.5% of the world's population aged 20-79, have diabetes, and 3 in 4 persons live in low- and middle-income countries (LMIC) (12). In South-East Asia alone, it is predicted that the number of adults with diabetes will reach 113 million and 151 million by 2030 and 2045, respectively (12).”

2. On page 6 under the Intervention overview, authors mentioned a previous qualitative study on line 157 but information has not been provided on how this is linked with the current study. It will be useful to provide this information

Thank you. We have added in line 164 - 167:

“In the qualitative study we identified obstacles, barriers and facilitations pertinent to having T2D and a strong desire for knowledge and peer support in self-management among people with T2D in rural communities (44, 45). This knowledge fed into the process of designing the diabetes management intervention, "Living healthy and well with diabetes,…."

3. Materials and Methods

Page. 10. Under sampling, 39 people in the pre-study refused to participate in the diabetes club intervention but mention has been made of this as a possible influencing factor of the outcome/results. Indicate the reasons why they refused to participate in the intervention. That may help the explanation.

Thank you for this very relevant question.

We do not have solid information on why 39 refused to participate in the diabetes club intervention, but we believe that there are a plethora of reasons. We have added to the discussion in line 445 - 453 :

“This pre-/-post-study aimed to assess the feasibility of a peer-based club intervention in influencing the HbA1c index and self-management among people living with T2D in rural communities. Albeit there were no effects on the primary outcome, the study still shows promising results that should be addressed in a randomized controlled study. A total of 39 persons refused to participate in the intervention. We have not systematically recorded the reasons for this nonparticipation, but believe that there is a plethora of reasons including logistics, reluctance to participate in new types of interventions, and the COVID-19 crisis probably also contributed to our intervention participants' dropout. Future studies should address how to reach out to these nonparticipants in order to increase participation.”

4. Results

Combine Table 4 and 5 into one table as these are the self-management practices

Combine table 6 and 7 into one table

We have combined table 4 and 5 into one table. However, regarding combing table 6 and 7 into one table was a challenge as the measures in is N in table 7 vs biological measures. And we feel that combing the table will make it more difficult for the reader to understand the table. 

5. Discussion

All the sub-sections under the discussion have been linked with the COVID-19 pandemic but there is not report on whether any of the participants suffered the COVID-19 It will be appropriate to mention this in order to put the discussion in the appropriate context as mentioned under the settings on page 7 line 145 - 154

For the discussion of the HbA1c changes, COVID-19 is used as a plausible reason for no significant difference but this did not seem to affect self-care practices including medication adherence. Self-care practices are known to have effect on the HbA1c levels. it will be useful to find explanations for the improvement in self-care practices but didn't reflect in the HbA1c

On page 22 line 432 - 434, a possible reason for increase in HbA1c is indicated as less fear of diabetes complication. Support this statement with evidence

We changed the discussion as proposed by the reviewer

The average HbA1C levels demonstrated an unexpected statistically significant though minimal and probably not clinically relevant increase from 6.9 to 7.24 % from pre-intervention to post-intervention. Our result contrasts with some previous studies in Cameroon (50), Uganda (25), and Guatemala (24), where peer support significantly improves HbA1c in people with T2D. One explanation for the slight increase in HbA1c was that our intervention was conducted during the delta variant outbreak of COVID-19 in Vietnam (43), not sparing Thai Binh Province. Albeit it is unknown how many of the participants were infected by COVID-19 but nationwide social distancing during COVID-19 (45-50) posed challenges like a more sedentary lifestyle and consequently inadequate diabetes self-management among people with T2D (21). Individuals with T2D also feared being more susceptible to COVID-19 infection due to comorbidity (21-23), leading to the delay or avoidance of regular check-ups at healthcare facilities (51-55). Further, we do not know the changes in HbA1C among patients who did not participate in our study. Another explanation for the small increase in HbA1c is that T2D is a progressive condition (56). In our study, the elderly predominated with 79.3% of respondents aged 60 and above; thus, although the normal patterns of development of T2D can be slowed down based on good self-management, they may still worsen with time. In addition, we might hypothesize that having less fear of diabetes complications may contribute to the small increase HbA1c index due to a slightly less adherence to recommendations. However, our data does not support this assumption and future analyses should look into this area. We also found similarities in the previous studies, which exhibited a non-significant difference in HbA1c level (29-31) after the intervention. Moreover, the number of studies with similar findings was limited, which may be caused by the fact that positive results tend to be published, while no results or negative tend to be unpublished (57).

And “Unfortunately the improvement in self-management behaviors was not reflected in improved Hb1Ac.”

---

## [Editor Report · Decision Letter 1]

7 Aug 2023

Pre-/-post-analyses of a  feasibility study of a peer-based club intervention among people living with type 2 diabetes in Vietnam’s rural communities

PONE-D-23-09542R1

Dear Dr. Dang,

We’re pleased to inform you that your manuscript has been judged scientifically suitable for publication and will be formally accepted for publication once it meets all outstanding technical requirements.

Kind regards,

Tahir Turk, PhD

Academic Editor

PLOS ONE
---

## [Editor Report · Acceptance letter]

6 Oct 2023

PONE-D-23-09542R1 

Pre-/-post-analyses of a  feasibility study of a peer-based club intervention among people living with type 2 diabetes in Vietnam’s rural communities 

Dear Dr. Dang:

I'm pleased to inform you that your manuscript has been deemed suitable for publication in PLOS ONE. Congratulations! Your manuscript is now with our production department. 

Kind regards, 

on behalf of

Dr. Tahir Turk 

Academic Editor

PLOS ONE